# A New Chemotactic Mechanism Governs Long-Range Angiogenesis Induced by Patching an Arterial Graft into a Vein

**DOI:** 10.3390/ijms231911208

**Published:** 2022-09-23

**Authors:** Dhisa Minerva, Nuha Loling Othman, Takashi Nakazawa, Yukinobu Ito, Makoto Yoshida, Akiteru Goto, Takashi Suzuki

**Affiliations:** 1Center for Mathematical Modeling and Data Science, Osaka University, 1-3 Machikaneyama-cho, Toyonaka City 560-8531, Japan; 2Faculty of Computer Science and Information Technology, Universiti Malaysia Sarawak, Jalan Datuk Mohammad Musa, Kota Samarahan 93400, Malaysia; 3Department of Cellular and Organ Pathology, Graduate School of Medicine, Akita University, 1-1-1 Hondo, Akita City 010-8543, Japan

**Keywords:** angiogenesis, chemotaxis, arterial patch, hybrid simulation, cell driving force, VEGF

## Abstract

Chemotaxis, the migration of cells in response to chemical stimulus, is an important concept in the angiogenesis model. In most angiogenesis models, chemotaxis is defined as the migration of a sprout tip in response to the upgradient of the VEGF (vascular endothelial growth factor). However, we found that angiogenesis induced by performing arterial patch grafting on rabbits occurred under the decreasing VEGFA gradient. Data show that the VEGFA concentration peaked at approximately 0.3 to 0.5 cm away from the arterial patch and decreased as the measurement approaches the patch. We also observed that the new blood vessels formed are twisted and congested in some areas, in a distinguishable manner from non-pathological blood vessels. To explain these observations, we developed a mathematical model and compared the results from numerical simulations with the experimental data. We introduced a new chemotactic velocity using the temporal change in the chemoattractant gradient to govern the sprout tip migration. We performed a hybrid simulation to illustrate the growth of new vessels. Results indicated the speed of growth of new vessels oscillated before reaching the periphery of the arterial patch. Crowded and congested blood vessel formation was observed during numerical simulations. Thus, our numerical simulation results agreed with the experimental data.

## 1. Introduction

Angiogenesis is the formation of new blood vessels that occurs under the control of chemical signals in the body. It is responsible for a wide variety of physio and pathological processes and the progression of diseases such as cancer, cardiomyocytes, or arteriovenous malformations (AVMs). Vascular endothelial growth factor (VEGF) is one of the chemical signals that bind to receptors on the surface of endothelial cells [1]. When VEGF binds to the surface receptors, endothelial cells become activated and detached from the blood vessel and subsequently migrate to create new blood vessels. The migration of new blood vessels is regulated by the spatial gradient of the signal. This process is called chemotaxis and the signal is called a chemoattractant. The endothelial cell at the front of the sprout tip senses the VEGF gradient to lead the direction of migration. This migration is then followed by other endothelial cells that then create a path line of a new blood vessel. Typically, in angiogenesis, chemical signals are released at the time and place where they are required. Therefore, new blood vessel formation is well-balanced in processes such as growth and healing. However, in the state referred to as the angiogenic switch [2], the chemical signals can occasionally become unbalanced and irregular, causing abnormalities in the new blood vessel formation, leading to vascular anomalies. For example, arteriovenous fistulas (AVFs) and arteriovenous malformations (AVMs).

Arteriovenous fistulas (AVFs) are the anomalies in which arteries and veins are directly connected, bypassing the capillary network that results in the shunting of blood between the two. AVFs are distinct from AVMs and are congenital lesions that arise due to failure of embryonic or fetal vascular differentiation and are also related to noniatrogenic trauma [3,4]. Usually, they increase in size if untreated. As for arteriovenous malformation (AVM), it is an abnormal angiogenesis consisting of feeder artery and drainage vein vessels which is abnormal, complex, and called “Nidus”. AVMs probably have high flow compared to AVFs, higher resistance, and may remain asymptotic [3]. AVMs can occur in organs of the human body such as the brain, lungs, intestinal tract, and the spinal dura mater [5,6,7,8]. So far, the cause of AVMs is not entirely clear. In particular, the process of AVF formation, which is a crucial step in AVM, pathogenesis is not yet understood. Moreover, the role of angiogenic factors, including the VEGF, in AVMs is still unknown owing to the lack of spontaneous in vivo AVM models. In fact, the lack of in vivo AVM models affected the development of therapeutic medications for AVMs [9].

Recently, ref. [10] have successfully induced AVF by patching an arterial graft into a rabbit’s vein. The new blood vessels were distinguishable from the pre-existing linear blood vessels owing to strong flexion and significant meandering. In this study, we extend the study in [10] by analyzing the VEGF concentration during AVF formation in an arterial patch procedure. The VEGF concentration was measured at a certain period of time. We observed that the VEGF concentration oscillated and gradually attenuated with time. Based on this data, we developed an angiogenesis model by introducing a new chemotactic velocity that governs new blood vessel formation under certain VEGF concentration changes.

Angiogenesis has been modeled using a variety of approaches including the Cellular Potts model [11,12,13], stochastic differential equations [14,15,16], compartment-based models [17], continuum modelling [18,19], and alternative mechanistic approach [20]; see [21,22] for recent reviews.

In this study, we took a hybrid approach and modified Anderson and Chaplain’s model [23]. In their model, it was assumed that the migration of sprout tip is influenced by random motility, chemotaxis in response to signal gradients, and haptotaxis in response to fibronectin gradients. As shown in the following equation, three factors contribute to the endothelial cell flux $J_{n}$: (1)$$J_{n}=J_{random}+J_{chemo}+J_{hapto},$$

They described the chemotactic flux $J_{chemo}$ as
(2)$$J_{chemo}=\chi n\nabla c,$$
where *n*, *c*, and χ denote the sprout tip density, chemoattractant concentration, and chemotactic coefficient, respectively. Generally, the conservation equation for the sprout tip density is in the form of
(3)$$n_{t}=-\nabla \cdot J_{n}.$$

In Section 2.5, we expressed spatiotemporal molecular concentrations as partial differential equations and individual cell movements as biased random walks. We found that our simulations were consistent with the experimental observations.

## 2. Results

### 2.1. VEGF Distribution Model

VEGF is the key mediator for angiogenesis and has been discovered to be a permeability-enhancing agent, leading to the disruption of intercellular contacts and increase of permeability [24]. The VEGF family of genes consists of at least seven various subtypes such as VEGFA, VEGFB and VEGFC, VEGFD, PIGF, VEGFE (Orf-VEGF), and trimeresurus flavoviridis svVEGF. However, VEGFA and its receptors, vascular endothelial growth factor receptor (VEGFR), including VEGFR-1 and VEGFR-2, play a major role in pathological angiogenesis. VEGFA has a variety of functions such as vascular permeability activity and stimulation of cell migration. Here, VEGF is also known as VEGFA [25].

### 2.2. VEGFA Quantification in In Vivo Angiogenesis Model

The in vivo angiogenesis-inducing model in [10] sutures an arterial graft into the wall of the left common jugular vein in male rabbits for the initiation of angiogenesis. Using this model, we quantified the spatiotemporal VEGFA concentration with an antibody-based measurement method called ELISA. Detailed experimental protocols are given in Section 4.2. In this paper, we consider the VEGFA concentration as the representative of the VEGF concentration and use it as the basis for our mathematical modeling.

The VEGFA concentration at the arterial patch graft was low and the level was equal to that of the serum. The highest level of VEGFA concentration was observed near the pre-existing artery (F4/5). At any time point, VEGFA concentration was higher in the pre-existing artery (F4/5) than in the arterial patch graft (Fp) (see Figure 1b). VEGFA concentration also revealed a repeated increase and decrease in addition to gradual attenuation, where the approximate curve shows a sine wave that attenuates gradually (see Equation (Equation 4) below). In this angiogenesis-inducing model, angiogenesis originates from the second branch of the left subclavian artery, heading for the arterial graft in the common jugular vein. From these observations, it is presumed that VEGFA is primarily released from the area where new blood vessels sprout and grow. In other words, blood vessels migrate toward the VEGFA with low gradient concentration.

### 2.3. VEGFA Distribution Equation Based on In Vivo Quantification Data

Let c(x,t) be the distribution of VEGFA concentration at position x∈[0,1] or ${\left[0,1\right]}^{2}$ and time 0≤t≤T. Here, the VEGFA distribution equation is based on the VEGFA quantification of the arterial patch grafting in Section 2.2. Based on the quantification above, we assume the following conditions for the VEGFA distribution:1.The VEGFA concentration near the pre-existing vessel is always higher than that near the arterial patch (Figure 1a).2.The VEGFA concentration changes according to a sine wave that attenuates gradually, as in Figure 1b.

Define the distribution of VEGFA concentration c(x,t) as follows: (4)$$c\left(x,t\right)=\left\{\begin{array}{ccc} a\left\{b+\sin(2\pi \sigma t)\exp(-\gamma t)\right\}\left\{1-\frac{{\left(3-r_{p}\right)}^{2}}{\epsilon^{2}}\right\}, & \; & r_{p}>0.1\\ 0, & \; & r_{p}\leq 0.1 \end{array}\right.$$
where *a*, *b*, σ, γ, and ϵ are constants. Here, $r_{p}$ is the distance function from the outer side of the arterial patch, which is defined as
(5)$$r_{p}=\sqrt{{\left(x-x_{p}\right)}^{2}+{\left(y-y_{p}\right)}^{2}}$$
where $\left(x_{p},y_{p}\right)$ is the location of the patch center. We assumed that the arterial patch has a radius of 0.1 (dimensionless) and was placed on the left side of the boundary (i.e., $x_{p}=0$).

Figure 2b shows the modeled time course of VEGFA concentration c(x,t) near the pre-existing artery (x=1). Figure 2c shows the initial VEGFA distribution c(x,t)(t=0) at domain $\left(x,y\right)\in {\left[0,1\right]}^{2}$. We used this VEGFA profile for a numerical simulation of the angiogenesis model.

### 2.4. New Blood Vessel Length Measurement in In Vivo Angiogenesis Model

Here, we measured the new blood vessels with anomalous vessel walls sprouting from pre-existing arterioles (see Figure 3a). As shown in Figure 3b, the lengths of the new blood vessels increased gradually until day 3, while only a small change of length was observed from day 3 to 5. On day 6, the vessels grew again, and the growth continued until day 9. The new blood vessels did not grow much further after day 9. In this model, the arteriovenous fistula was formed between days 10 to 14 [10]. Therefore, we speculate that the angiogenesis process stopped once the new blood vessels that sprouted from the arterioles reached the arterial graft on day 9.

### 2.5. Sprout Tip Distribution Equation with New Chemotactic Velocity

We assumed that the migration of the sprout tip was influenced by random motility and chemotaxis in response to temporal changes in the VEGFA gradient, rather than the gradient itself. To derive the sprout tip distribution governed by these two factors, we considered the following mass conservation of sprout tip density n=n(x,t): (6)$$n_{t}=-\nabla \cdot \left(-d\nabla n+nv_{m}\right),$$
where *d* denotes the random motility constant and $v_{m}=v_{m}\left(x,t\right)$ is the transport velocity [23]. Equation (Equation 6) is divided into two terms. The first term explains random motility. The second term explains the migration of the sprout tip under the influence of a flow with velocity $v_{m}$. Partly inspired by the analysis of the LEGI (local excitation, global inhibition) model of chemotaxis (see Appendix B), we define
(7)$$v_{m}=\nabla f,$$
where
(8)$$f=\left\{\begin{array}{ccc} -\beta c_{t}, & \; & c_{t}>0\\ 0, & \; & c_{t}\leq 0. \end{array}\right.$$

Here, β is constant and $c_{t}$ in (Equation 8) is the derivative of our VEGFA distributions (Equation 4) and (Equation 5).

The application of (Equation 8) was also inspired by the comparison of the growth of new blood vessels (Figure 3b) and the approximation curve of VEGFA concentrations (Figure 1c). The growth speed of new blood vessels was relatively fast from day 0 to 2 and from day 5 to 8, which seems to coincide with the timing when the VEGF concentrations increased near the pre-existing vessel. However, we mentioned that we cannot decisively say the above two must be correlated, owing to relatively high variability in the vessel growth data (Figure 3b).

## 3. Discussion

Here we used the temporal change in the chemoattractant gradient as the new chemotactic mechanism to model the arterial patch-induced angiogenesis. The temporal changes in VEGF concentration affected the angiogenesis process. This was probably caused by the hypoxia state around the arterial patch, which affected the VEGF attenuates status to stimulate the sprout tip of the endothelial cells. Our angiogenesis model successfully predicted new blood vessel growth that fitted the experimental data. During the numerical simulation, we successfully obtained twisted and congested formations that were formed from t=2 to t=7. This formation was pathological blood vessels characteristic.

We compared the length of the new blood vessel at each time during the numerical simulation, from t=1 to t=14 with the experimental results [10], as shown in Figure 5. Curves in different colors show different simulation results under the same parameter sets (see Figure 6b). Although we used dimensionless parameters for the numerical simulation, the simulation results showed patterns that were similar to those from in vivo measurements (see Figure 6). Because of the lack of experimental data and little information, we can see a differentiation value in error for the mean of each new blood vessel’s growth from in vivo. The growth of the new blood vessels was initially fast before decelerating at some time period. Subsequently, the new blood vessels regrow until reaching the arterial patch. Additionally, we also ran our simulations with different parameters since our method also depends on spatial grid step discretization (see Appendix A).

## 4. Material and Methods

### 4.1. Numerical Simulation Method and Results

#### 4.1.1. Derivation of Numerical Scheme

To understand our angiogenesis model, we solve systems (Equation 4)–(Equation 8) numerically and perform a simulation of the new blood vessel growth using a hybrid technique. This technique was invented by Anderson and Chaplain to simulate a tumor-induced angiogenesis model [23]. We herein generalize the hybrid technique such that the scheme can be used to simulate our systems (Equation 4)–(Equation 8) for any transport velocity $v_{m}$.

We consider the following system: (9)$$\left\{\begin{array}{c} \begin{array}{cccc} n_{t} & =-\nabla \cdot F, & \; & \mathrm{in}\;\;\Omega \times (0,T)\\ F= & -d\nabla n+G, & G=nv_{m} & \\ v_{m} & =\nabla f & \; & \\ f & =f(x,t) & \mathrm{in}\;\;(8) & \\ 0 & =\nu \cdot f & \; & \mathrm{on}\;\;\partial \Omega \times (0,T)\\ {n|}_{t=0} & =n_{0}\left(x\right)>0 & \; & \mathrm{on}\;\;\bar{\Omega } \end{array} \end{array}\right.$$
and define the domain of system (Equation 9) as Ω=(0,1)×(0,1).

The following numerical scheme is known to ensure both positivity and mass conservation of the solution [26]. The domain Ω is partitioned into two types of lattices, i.e., main and sub: (10)$$\begin{array}{ccc} \mathrm{Main}\;\mathrm{lattice}: & \left(x_{i},y_{i}\right)=\left(\left(i-\frac{1}{2}\right)h,\left(j-\frac{1}{2}\right)h\right), & i,j=1,\ldots ,N,\\ \mathrm{Sub}\;\mathrm{lattice}: & \left({\hat{x}}_{i},{\hat{y}}_{i}\right)=\left(ih,jh\right) & i,j=1,\ldots ,N, \end{array}$$
where h=1/N (see Figure 4a). For the time step, we renew the time increment, $\tau_{k}$, at every step *k*.
(11)$$t_{0},\;\;t_{k+1}=t_{k}+\tau_{k},\;\;\;\;\;\;k=0,\ldots ,M-1,\;\;\;\;\;\;t_{k}\leq T,$$
where $\tau_{k}$ will be defined later.

In these lattices, we assume that *n* and $v_{m}$ are approximated on the main lattice. Additionally, we assume that *f*, *G*, and *F* are approximated on the sub-lattice: (12)$$\begin{array}{cc} n_{i,j}^{k}\approx n\left(x_{i},y_{i},t_{k}\right), & {\left(v_{m}\right)}_{i,j}^{k}\approx v_{m}\left(x_{i},y_{j},t_{k}\right),\\ G_{i,j}^{k}\approx G\left(x_{i},y_{j},t_{k}\right), & F_{i,j}^{k}\approx F\left({\hat{x}}_{i},{\hat{y}}_{j},t_{k}\right). \end{array}$$

Here, ${\left(v_{m}\right)}_{i,j}^{k}=\left({\left(v_{m}\right)}_{i,j,1}^{k},{\left(v_{m}\right)}_{i,j,2}^{k}\right)$ is calculated on the *x* and *y* directions using the average of four points surrounding every (1≤i,j≤N).
(13)$$\begin{array}{cc} {\left(v_{m}\right)}_{i,j,1}^{k} & =-\frac{\beta }{2}\left(\frac{f_{i,j}^{k}-f_{i-1,j}^{k}}{h}+\frac{f_{i,j-1}^{k}-f_{i-1,j-1}^{k}}{h}\right)\\ {\left(v_{m}\right)}_{i,j,2}^{k} & =-\frac{\beta }{2}\left(\frac{f_{i,j}^{k}-f_{i,j-1}^{k}}{h}+\frac{f_{i-1,j}^{k}-f_{i-1,j-1}^{k}}{h}\right) \end{array}$$
where $f_{i,j}^{k}=f\left({\hat{x}}_{i}^{k}\right)$, and *f* is calculated on the sub-lattices. For f(x,t) in (Equation 8), ${\left(c_{t}\right)}_{i,j}^{k}$ can be calculated by taking the time derivative of (Equation 8) on the sub-lattices, ${\left(c_{t}\right)}_{i,j}^{k}=c_{t}\left({\hat{x}}_{i},{\hat{y}}_{j},t_{k}\right)$.

Define $v_{m}$ as
(14)$${\left(v_{m}\right)}_{i,j,1}^{k,\pm }=\max\left\{0,\pm {\left(v_{m}\right)}_{i,j,1}^{k}\right\},\;{\left(v_{m}\right)}_{i,j,2}^{k,\pm }=\max\left\{0,\pm {\left(v_{m}\right)}_{i,j,2}^{k}\right\}$$
for all (1≤i,j≤N) as the direction setting of the upwind side.

Next, we consider the sub-lattice where $G_{i,}^{k}$ is defined. In the *x* direction, we assume that $n_{i,j}^{k}$ and $n_{i+1,j}^{k}$ are brought to $G_{i,j}^{k}$ by flows ${\left(v_{m}\right)}_{i,j,1}^{k,+}$ and ${\left(v_{m}\right)}_{i+1,j,1}^{k,-}$, respectively. We may also assume that $n_{i,j}^{k}$ and $n_{i,j+1}^{k}$ are brought to $G_{i,j}^{k}$ by flows ${\left(v_{m}\right)}_{i,j,2}^{k,+}$ and ${\left(v_{m}\right)}_{i,j+1,2}^{k,-}$, respectively, in the *y* direction; see Figure 4b. Thus, we write $G_{i,j}^{k}=\left(G_{i,j,1}^{k},G_{i,j,2}^{k}\right)$ as follows:(15)$$\begin{array}{cc} G_{i,j,1}^{k} & =n_{i,j}^{k}{\left(v_{m}\right)}_{i,j,1}^{k,+}-n_{i+1,j}^{k}{\left(v_{m}\right)}_{i+1,j,1}^{k,-},\;\left(1\leq i\leq N-1,1\leq j\leq N\right),\\ G_{i,j,2}^{k} & =n_{i,j}^{k}{\left(v_{m}\right)}_{i,j,2}^{k,+}-n_{i,j+1}^{k}{\left(v_{m}\right)}_{i,j+1,2}^{k,-},\;\left(1\leq i\leq N,1\leq j\leq N-1\right). \end{array}$$

Therefore, $F_{i,j}^{k}=\left(F_{i,j,1}^{k},F_{i,j,2}^{k}\right)$ can be approximated by the following scheme.
(16)$$\begin{array}{cc} F_{i,j,1}^{k} & =-d\frac{n_{i+1,j}^{k}-n_{i,j}^{k}}{h}+G_{i,j,1}^{k},\;\left(1\leq i\leq N-1,1\leq j\leq N\right),\\ F_{i,j,2}^{k} & =-d\frac{n_{i,j+1}^{k}-n_{i,j}^{k}}{h}+G_{i,j,2}^{k},\;\left(1\leq i\leq N,1\leq j\leq N-1\right). \end{array}$$

By the zero-flux in Equation (Equation 5) of a system (Equation 9), we have the following conditions: (17)$$F_{0,j,1}^{k}=F_{N,j,1}^{k}=F_{i,0,2}^{k}=F_{i,N,2}^{k}=0\;\left(0\leq i,j\leq N\right).$$

Hence, $n_{i,j}^{k}$ is approximated by the following numerical scheme: (18)$$\frac{n_{i,j}^{k+1}-n_{i,j}^{k}}{\tau_{k}}=-\left(\frac{F_{i,j,1}^{k}-F_{i-1,j,1}^{k}}{h}+\frac{F_{i,j,2}^{k}-F_{i,j-1,2}^{k}}{h}\right).$$

We substituted (Equation 13), (Equation 15), and (Equation 16) into (Equation 18), which results in the following scheme: (19)$$n_{i,j}^{k+1}=n_{i,j}^{k}P_{i,j}^{k,0}+n_{i+1,j}^{k}P_{i+1,j}^{k+1}+n_{i-1,j}^{k}P_{i-1,j}^{k,2}+n_{i,j+1}^{k}P_{i,j+1}^{k,3}+n_{i,j-1}^{k}P_{i,j-1}^{k,4},$$
where
(20)$$\begin{array}{cc} P_{i,j}^{k,0} & =1-4\frac{d\tau_{k}}{h^{2}}-\frac{\tau_{k}}{h}\left({\left(v_{m}\right)}_{i,j,1}^{k,+}+{\left(v_{m}\right)}_{i,j,1}^{k,-}+{\left(v_{m}\right)}_{i,j,2}^{k,+}+{\left(v_{m}\right)}_{i,j,2}^{k,-}\right) \end{array}$$
(21)$$\begin{array}{cc} P_{i+1,j}^{k,1} & =\frac{d\tau_{k}}{h^{2}}+\frac{\tau_{k}}{h}{\left(v_{m}\right)}_{i+1,j,1}^{k,-}, \end{array}$$
(22)$$\begin{array}{cc} P_{i-1,j}^{k,2} & =\frac{d\tau_{k}}{h^{2}}+\frac{\tau_{k}}{h}{\left(v_{m}\right)}_{i-1,j,1}^{k,+}, \end{array}$$
(23)$$\begin{array}{cc} P_{1,j+1}^{k,3} & =\frac{d\tau_{k}}{h^{2}}+\frac{\tau_{k}}{h}{\left(v_{m}\right)}_{i,j+1,2}^{k,-}, \end{array}$$
(24)$$\begin{array}{cc} P_{i,j-1}^{k,4} & =\frac{d\tau_{k}}{h^{2}}+\frac{\tau_{k}}{h}{\left(v_{m}\right)}_{i,j-1,2}^{k,+} \end{array}$$
for (2≤i,j,≤N−1). The same procedure also applied for i,j=1…N to obtain the scheme at the boundary. From the scheme above, the values of $P_{i,j}^{k,0}$, $P_{i+1,j}^{k,1}$, $P_{i-1}^{k,2}$, $P_{i,j+1}^{k,3}$, and $P_{i,j-1}^{k,4}$ are the weights from each point of (i,j)’s neighbors to approximate the value of $n_{i,j}^{k+1}$. Assume five *P* values as the probability of a cell remaining at (i,j), migrating to the left (i−1,j), to the right (i+1,j), to the downside (i,j−1), and to the upside (i,j+1), respectively, if we shift $P_{i+1,j}^{k,1}$, $P_{i-1,j}^{k,2}$, $P_{i,j+1}^{k,3}$, and $P_{i,j-1}^{k,4}$ to $P_{i,j}^{k,1}$, $P_{i,j}^{k,2}$, $P_{i,j}^{k,3}$, and $P_{i,j}^{k,4}$, respectively, see Figure 4d.

During angiogenesis, a new sprout tip can be generated from the existing sprout tips. Additionally, the sprout tips can form loops by connecting with other sprout tips or the existing new blood vessels. These two events are called branching and anastomosis, respectively [27]. We adopted the branching and anastomosis rules from the discrete mathematical model of tumor-induced angiogenesis [23]. Namely, for the branching to occur, the lifetime of sprouts must exceed the threshold age $T_{branching}$, and the branching probability is given by
$$P_{branching}=\left\{\begin{array}{ccc} 0, & 0\leq c<0.25, & \\ 0.3, & 0.25\leq c<0.45, & \\ 0.4, & 0.45\leq c<0.60, & \\ 0.5, & 0.60\leq c<0.70, & \\ 1, & 0.70\leq c<1.00 \end{array}\right.$$
with a modification that the branching probability is applied only when the temporal change in the VEGFA concentration is positive, which follows (Equation 8). When anastomosis occurs, only one sprout is allowed to continue to grow.

As detailed in [26], this numerical scheme is stable when the time increment $\tau_{k}$ is updated by the following criterion: (25)$$\tau_{k}\leq \frac{h^{2}}{4(d+h{\hat{\left(v_{m}\right)}}^{k})},$$
where
(26)$${\hat{\left(v_{m}\right)}}^{k}=\max_{1\leq i,j\leq N}\left\{{\left(v_{m}\right)}_{i,j,1}^{k,\pm },{\left(v_{m}\right)}_{i,j,2}^{k,\pm }\right\},$$
satisfied
(27)$$\sum_{i=0}^{4}P_{i,j}^{k,l}=1$$
and
(28)$$P_{i,j}^{k,l}\geq 0\;\;\left(l=0,1,2,3,4\right),$$
which confirms the probability assumptions of *P*s from schemes (Equation 19)–(Equation 24) is reasonable.

#### 4.1.2. Numerical Simulation Results

In this section, the domain setting and parameter values used in the simulations are depicted. All the parameters are dimensionless. As discussed in Section 2.3, we assumed that an arterial patch of radius 0.1 (dimensionless) was placed in the center of the left boundary. Thus, we set $r_{p}=\sqrt{x^{2}+{\left(y-0.5\right)}^{2}}$. Initially, we placed 20 sprout tips randomly at the right boundary. This position can be considered to be that of the pre-existing vessel (see Figure 1a and Figure 2a) by setting the length of domain-*x* as the actual in vivo length. From the in vivo measurement, the distance between the arterial patch and the pre-existing vessel is varied (0.3±0.2 cm). The other parameters we used in this numerical simulation are d=0.0001728, β=1, a=1, b=0.39, σ=0.23/2, γ=0.15, and ϵ=2.9 (see Section 2.3).

We show our numerical simulation results of the angiogenesis model (Equation 4)–(Equation 8) in Figure 5, Figure 6 and Figure 7, respectively. Figure 8 shows the initial state of the arterial patch on the left side of the boundary and pre-existing vessel on the right side of the boundary at t=0. The VEGF distribution is higher near the pre-existing vessel and lower towards the arterial patch.

We simulated for T=0 until T=14 and tracked down the growth of the new blood vessels at t=1,2,7,10,11, and 14. In these figures, the yellow to orange color in the background of each picture represents the VEGFA concentration (see also Equation (Equation 4) and Figure 8). In Figure 5, we see that at t=1 and t=2 the VEGFA concentration is higher (orange), but it gets lower (yellow) at later times. Since the VEGF concentration attenuates gradually (see Figure 9a) over time followed by the sprout tips of the newly generated blood vessels, we can see changes in color from orange to yellow and vice versa.

We looked at the blood vessel growth in Figure 5. At t=2, the growth of new blood vessels started to decelerate and the sprout tips moved randomly. The random movement of the sprout tips heightened the probability of the blood vessels meeting other tips which caused the congested and twisted formation. This creates an AVM formation such as the one in Figure 5c. Thus, many sprout tips died owing to the anastomosis. The fewer the sprout tips generated initially, the lower the probability of anastomosis. In fact, there were two sprouts at most (from 20 sprout tips generated initially) that survived and reached the arterial patch resulting in vascularization (see Figure 5d) during the numerical simulation.

Figure 6 shows temporal changes in the VEGF gradient causing the graphs to have a high and almost linear slope between in vivo experiments and our simulations. We ran a few simulations and calculated the vessel length in millimeters (mm). We can see that the two graphs are almost similar. Both graphs show that at day 2, the growth of the sprout tip started to decline slowly but regrow faster on day 7 until it reached the periphery of the arterial patch on day 10.

Data for each new blood vessels length from simulations and in vivo are presented in the graph (see Figure 7) of mean ± standard deviation (SD), 1.04571±0.47451 and 0.80±0.22234, respectively. Statistical analysis was evaluated by Student’s *t*-test and the *p* value was <0.05. Figure 7b shows that the result was statistically significant for each day. As shown in Figure 7b, we can see from day 2 until day 7 the growth speed of blood vessels becomes slow and almost constant and the speed becomes faster and higher after day 7 and become constant again starting from day 10. The attenuated status of VEGF also affected the growth speed of the blood vessels. The *p* value in vivo was >0.05, which is statistically not significant (see Figure 7a). However, our simulation results show similar patterns to those in vivo experiments (Figure 5).

### 4.2. Material and Methods: In Vivo

#### 4.2.1. Quantification of Vegfa in In Vivo Angiogenesis Model

Three rabbits at each time point (1, 3, 6, 12, 18 h, 1–7 days; different rabbits each time) after the grafting was euthanized, and the arterial graft and fatty tissue were collected and then divided into approximately eight areas, as shown schematically in Figure 1a. The total protein was extracted from the tissues by the Minute Detergent-Free Total Protein Extraction Kit obtained from Invent Biotechnologies, Inc. (Minneapolis, MN, USA). The level of VEGFA concentration in the tissue was measured by Rabbit VEGFA ELISA kits purchased from Cusabio Biotech Co., Ltd. (Wuhan, China). The concentration of VEGFA in each sample was calculated based on a recombinant protein-based standard curve (Appendix A). The Multiskan JX spectrophotometer (Thermo Fisher Scientific Inc., Yokohama, Japan) was used to measure the absorbance. All assays were performed in duplicate, and the mean values of the results were used in the analyses.

#### 4.2.2. Measurement of New Blood Vessel Length

Forty-five rabbits were subjected to the patch procedure. Three rabbits at each time point (control, 1–14 days after the patch procedure) were euthanized; the patch-bearing left the common jugular vein and its surrounding tissues were subjected to histological evaluation. The tissues were cut into six 5 mm thick serial slices in an approximately horizontal plane including the cross sections of the arterial patch and the second branch of the left subclavian artery, which were embedded in paraffin. The paraffin-embedded samples were subsequently cut into 4 μm thick sections and stained with Haematoxylin-Eosin and Elastica-Masson. Glass slides were converted into virtual slides using the NDP slide scanner (Nanozoomer 2.0HT) (Hamamatsu Photonics K.K., Shizuoka, Japan), a virtual microscopy system. Measurement of new blood vessel length was carried out using a distant measuring tool of the viewing platform (NDP.View2) (Hamamatsu Photonics K.K., Shizuoka, Japan). Six slices were observed per subject, and 5 to 10 new blood vessels were identified per slice.

#### 4.2.3. Numerical Simulation

Numerical simulation was performed in Python. The original codes are available at: https://github.com/rtminerva/hybrid-simulation/tree/master/angiogenesis/Arterial$%$20Patch$%$20Grafting, (accessed on 9 August 2020).

We used run_main_code.py to run the whole code and coef_setting.py to adjust parameters.

## 5. Conclusions

The process of AVM formation is still unknown. This is because human AVMs are rarely recognized before the onset of symptoms, and their condition can only be confirmed at the time of surgery or autopsy. Furthermore, since there is no animal model that accurately reproduces AVMs, it has been difficult to observe the formation process of AVMs and changes in angiogenic factors over time.

In this study, by observing the AVM animal model newly developed in [10] over time, we confirmed the cyclic variation of VEGF in the early stage of AVM formation. The mathematical model obtained from this study was able to predict the formation of AVMs owing to temporal changes in VEGF. The mathematical simulation reproduced the tangled neovascularization characteristic of AVMs. The simulated elongation process of neovascularization until the formation of arteriovenous fistula suggested that some neovascularization was discarded. This study is expected to shed more light on the early formation process of AVMs, which has not been observed so far.

In addition, in recent years, anti-VEGF monoclonal antibodies have been used clinically in cancer treatment and are widely administered to various carcinomas [28,29,30]. The results of this study suggest that cyclic variation of VEGF plays an important role in the formation of AVMs. Therefore, the administration of anti-VEGF monoclonal antibodies can be considered as a therapeutic agent for AVMs, and it is expected that the effective timing of administration can be predicted mathematically.

## Figures and Tables

**Figure 1 ijms-23-11208-f001:**
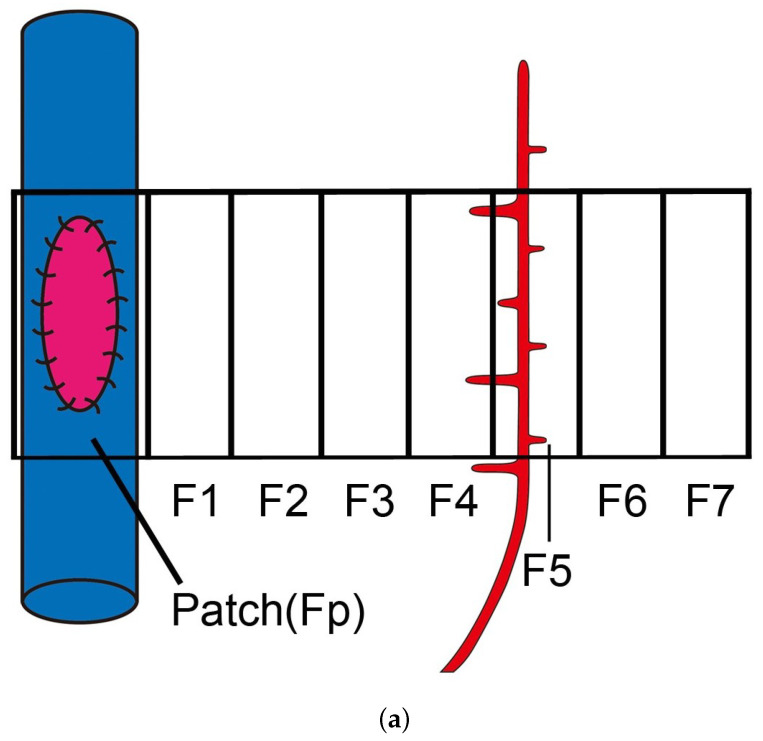
VEGF distribution at the tissue area and approximation curve of VEGF concentration from day 0 to 14 in in vivo experimental. (**a**) An illustration of the eight tissue areas (Fp, F1, F2, F3, F4, F5, F6, and F7) that have been taken to measure the VEGFA concentration. The arterial patch graft is placed at Fp and the pre-existing artery is assumed to exist at around F4/F5 areas. (**b**) VEGF concentration from experimental data showing that the VEGF concentration was higher in the pre-existing area (F4/5) than in the patch area (Fp). (**c**) The measured VEGF concentrations (pg/mL) at the pre-existing vessel change with time. This indicates that there were three rabbit cases for each time point. We plotted each case on the graph, and its fitted graph was derived. The curve was obtained by numerically fitting the data to Equation (Equation 4): a=46.2525, b=0.2131, σ=0.1548, γ=0.3485.

**Figure 2 ijms-23-11208-f002:**
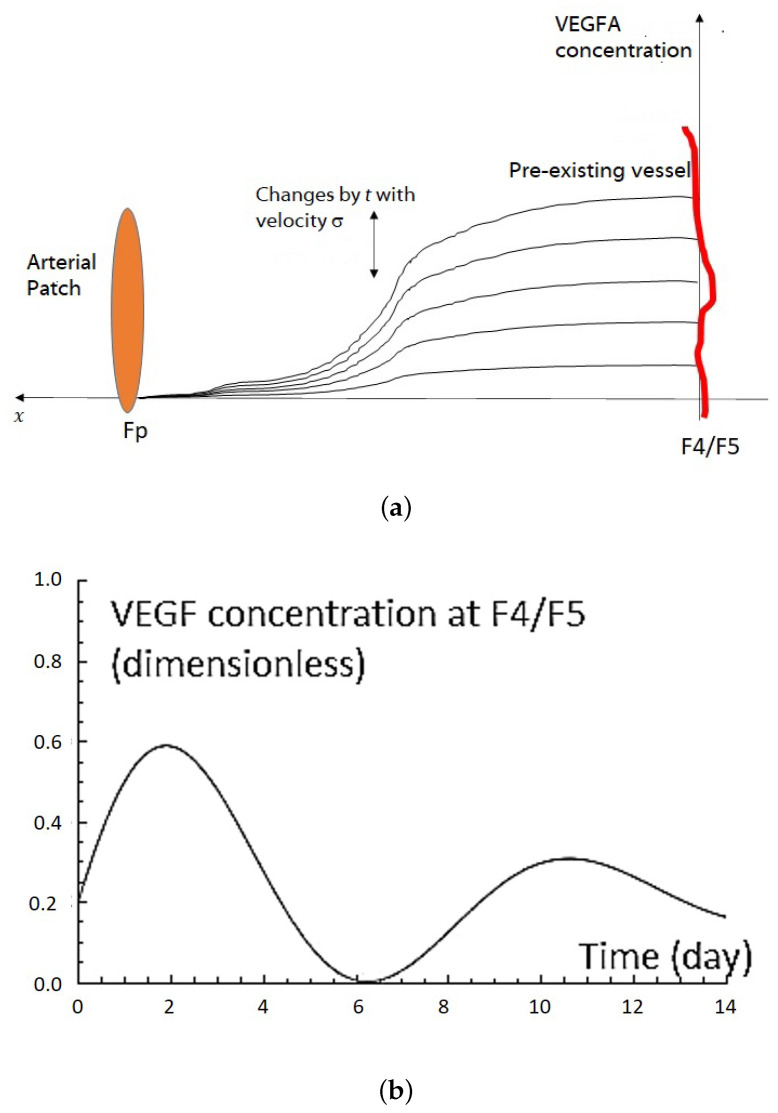
(**a**) An illustration of VEGFA concentration changes over time. The level concentration of VEGFA changes according to {b+sin(2πσt)exp(−γt)} term in Equation (Equation 4). The VEGFA concentration is always higher near the pre-existing blood vessels other than in the other area. This illustration is based on the experimental data in Section 2.2. (**b**) VEGFA concentration profile using Equations (Equation 4) and (Equation 5) at F4/F5 by setting parameters as follows: x,y∈(0,1), a=1, b=0.39, σ=0.23/2, γ=0.15, ϵ=2.9, $x_{p}=0$, $y_{p}=0.5$, x=1, and y=0.5. This VEGFA concentration changes graph has a qualitatively similar profile as that in vivo. The VEGFA concentration changes graph has a similar profile as in vivo experiments (see Figure 1b). (**c**) VEGF distribution of Equations (Equation 4) and (Equation 5) on domain x,y∈(0,1) at t=0.

**Figure 3 ijms-23-11208-f003:**
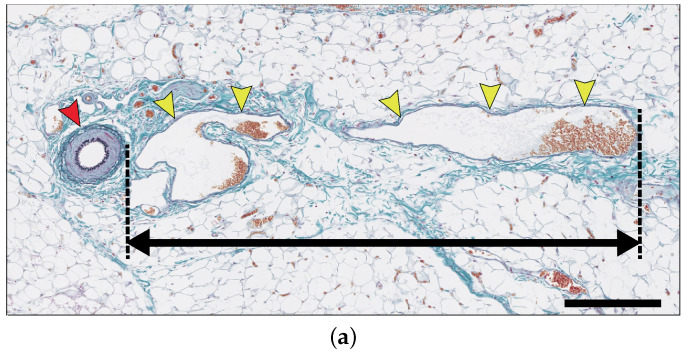
(**a**) New blood vessels (yellow arrowheads) are sprouting and growing from arterioles (red arrowheads). The measurement distance is between the outer circumference of the arteriole and the farthest point of the new blood vessel. The specimen was stained with Elastica-Masson staining. Scale bar: 250 μm. (**b**) Graph of the average length of the new blood vessel from day 0 to day 14. At each time point, every fifteen lengths were measured in three cases, and the average was calculated.

**Figure 4 ijms-23-11208-f004:**
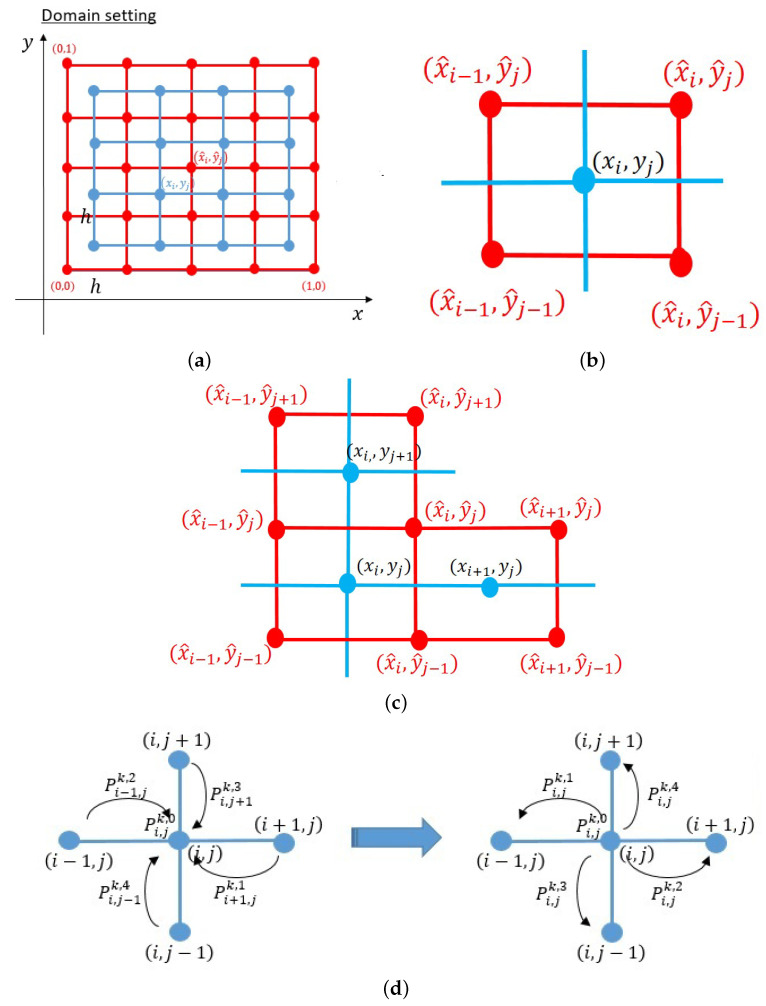
(**a**) The numerical lattices were defined on domain Ω=(0,1)×(0,1)). We defined two different lattices: the main lattice (blue) and sub-lattice (red). (**b**) Lattices for the calculation of scheme ${\left(v_{m}\right)}_{i,j}^{k}$. (**c**) Lattices for the calculation of scheme $G_{i,j}^{k}$. (**d**) We may shift the weight values $P_{i+1,j}^{k,1}$, $P_{i-1}^{k,2}$, $P_{i,j+1}^{k,3}$, and $P_{i,j-1}^{k,4}$ to $P_{i,j}^{k,1}$, $P_{i,j}^{k,2}$, $P_{i,j}^{k,3}$, and $P_{i,j}^{k,4}$, so that each of these *P* on (i,j) can be assumed as the probability of a sprout tip to migrate to the left, right, up, and bottom, respectively. Additionally, $P_{i,j}^{k,0}$ is regarded as the probability of a sprout tip to stay on (i,j).

**Figure 5 ijms-23-11208-f005:**
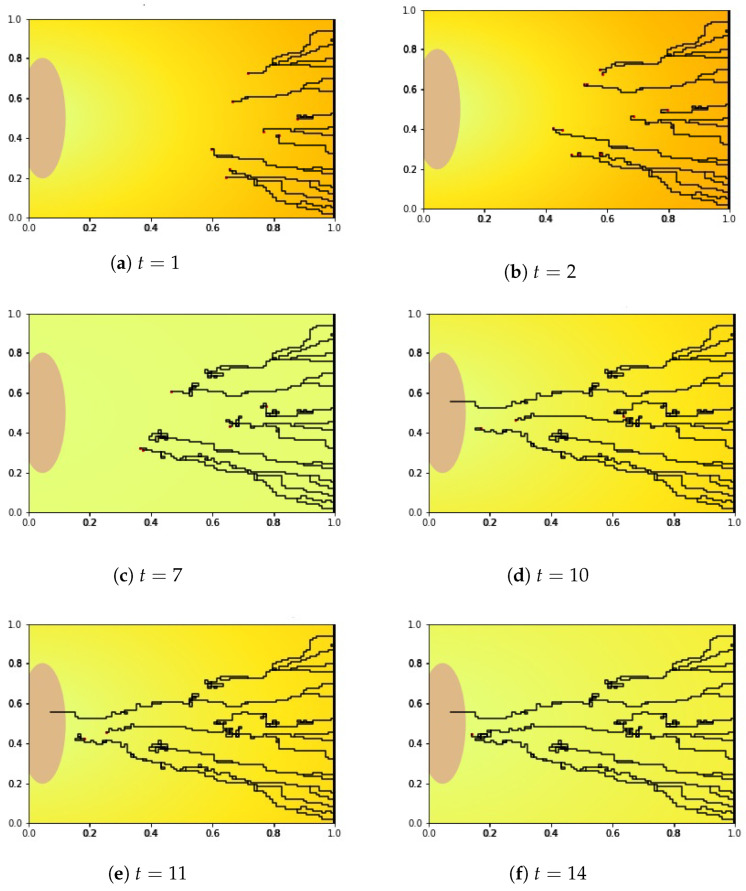
The simulation results of the new blood vessel growth with the background of VEGF concentration level, captured on days 1,2,7,10,11, and 14. Until t=2, the new blood vessels grew faster before the sprout tips experience the rest period. After t=7, the new blood vessels regrew until reaching the periphery of the arterial patch at t=10. Only one blood vessel can reach the arterial patch due to anastomosis.

**Figure 6 ijms-23-11208-f006:**
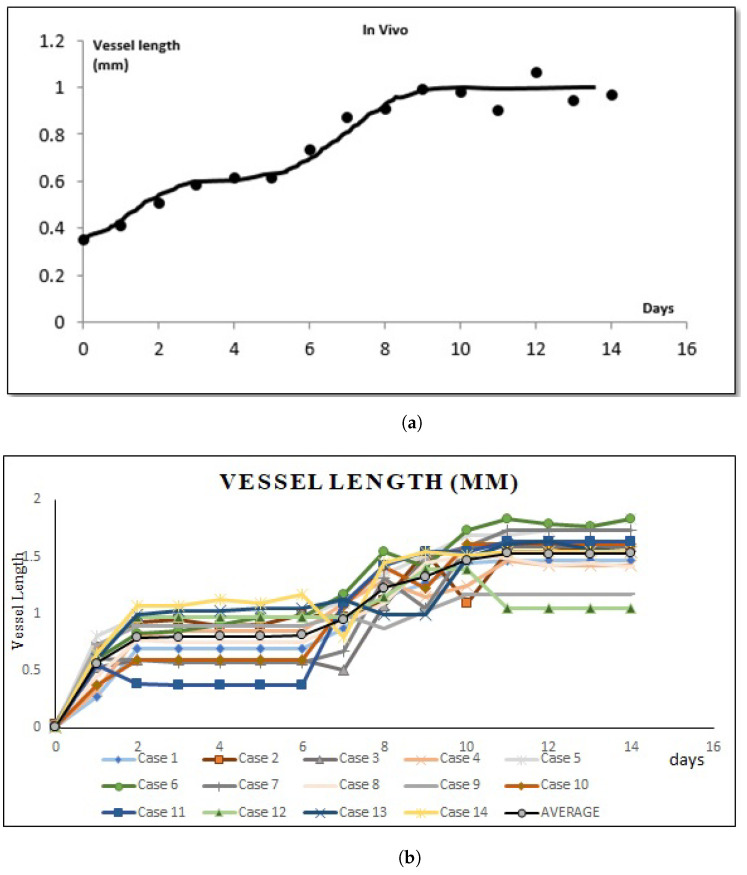
Average changes of new vessels’ length growth speed until it reaches the periphery of the arterial patch between in vivo and simulations from day 0 until day 14. (**a**) The average speed of vessel length growth in mm based on in vivo experiments. (**b**) The average speed of vessel length growth in mm based on simulations.

**Figure 7 ijms-23-11208-f007:**
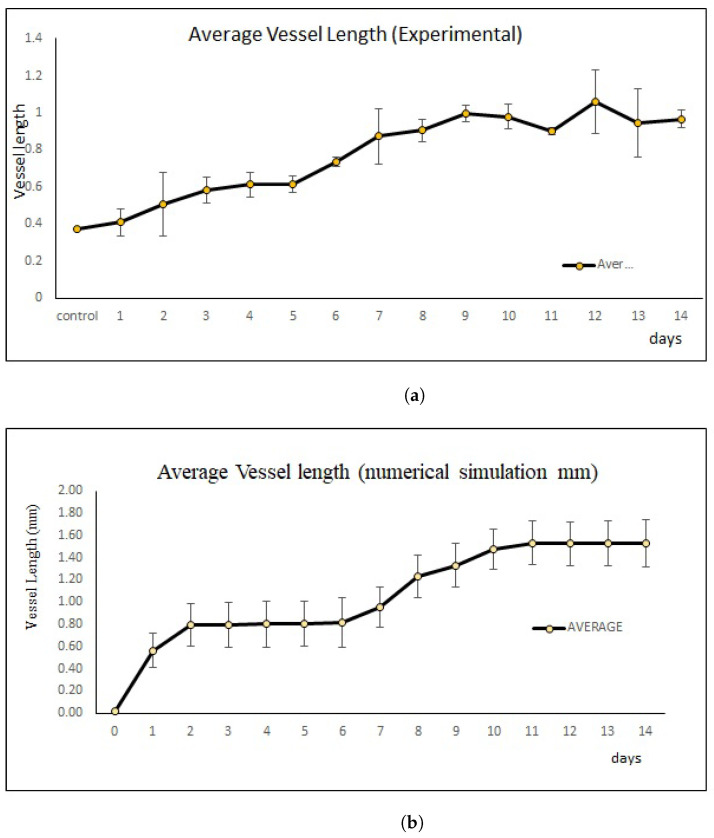
Error bars show the mean and standard deviation of the in vivo and simulations of the sprouting new blood vessels based on day 0 until day 14 until it reached the periphery of the arterial patch. (**a**) Mean and standard deviation of vessel length growth in mm based on in vivo experiments. (**b**) Mean and standard deviation of vessel length growth in mm based on simulations.

**Figure 8 ijms-23-11208-f008:**
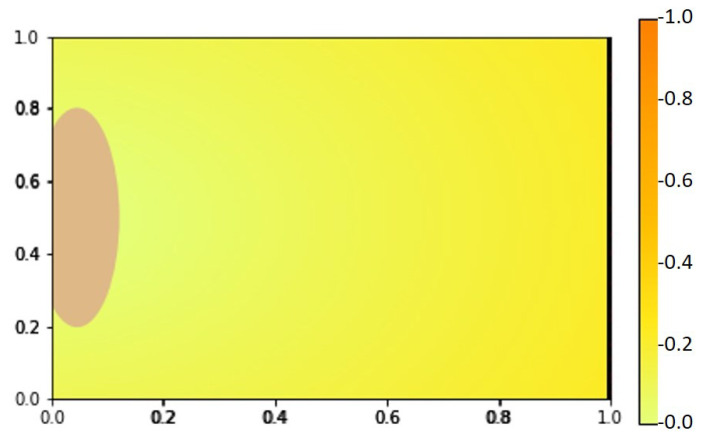
VEGF distribution at t=0 with the pre-existing vessel on the right and arterial patch on the left boundary.

**Figure 9 ijms-23-11208-f009:**
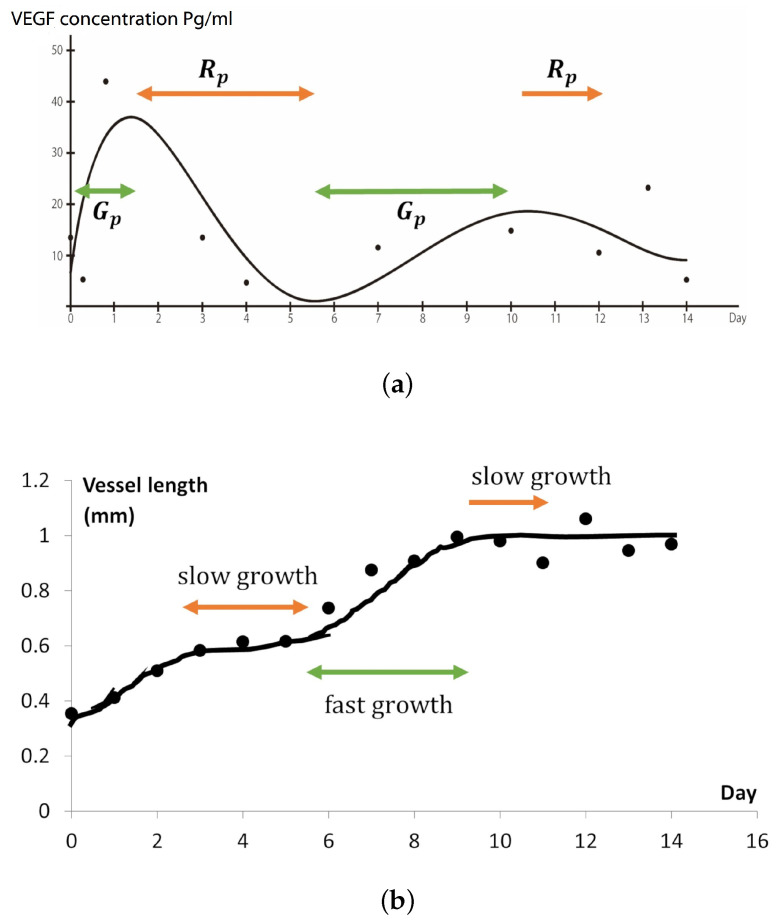
Comparison of the new blood vessel growth and the VEGFA gradient changes. When the VEGFA concentration is decreasing over time, the growth of the new vessels become slower. We consider this growth of the new blood vessels at the decreasing rate of VEGFA to be called the rest period, $R_{p}$, and at the increasing rate of VEGFA is called the growth period, $G_{p}$. (**a**) VEGFA concentration changing over time (days) and attenuates gradually. (**b**) Measurement of the new blood vessel in days.

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
