# Peer review of "A New Chemotactic Mechanism Governs Long-Range Angiogenesis Induced by Patching an Arterial Graft into a Vein"

_ijms, 2022, doi:10.3390/ijms231911208_

Round 1

Reviewer 1 Report

The manuscript is very interesting since it describes a mathematical model that relates VEGF gradients with the speed of vessels growth in a arterial patch. 

My comments for the authors are:

1. In Figure 1, in certain cases, for the same abscissa value the plot shows three values of ordinate. Why? 

2. The mathematical model  distinguished VEGF concentration near the pre-existing vessels from the VEGF concentration near arterial patch. The authors claim that VEGF concentration is higher in pre-existing vessels than in arterial patch. Has this model been validated with experimental data? The patch inclusion can disturb the homeostasis of endothelial cells and causes VEGF release. The authors considered that?

VEGF is a well-known angiogenic promoter, however, in future, it will be interesting to study others angiogenesis-related genes (TGF-beta, HIF-1alpha, FGF, among others). 

Reviewer 2 Report

Figure 2 a) letters to small to be able to read it.

Figure 3 a) needs to specify wich staining they are using. Figure 3 b) They are talking about average graph but did not specify how many samples they have checked. This need to be specified.

Round 2

Reviewer 1 Report

I recommend publishing the manuscript. 
